# Bone Conduction Cervical Vestibular Evoked Myogenic Potentials as an Alternative in Children with Middle Ear Effusion

**DOI:** 10.3390/jcm12196348

**Published:** 2023-10-03

**Authors:** Maxime Damien, Sylvette R. Wiener-Vacher, Pierre Reynard, Hung Thai-Van

**Affiliations:** 1Service d’Audiologie & Explorations Oto-Neurologiques, Hospices Civils de Lyon, 69003 Lyon, France; maxime.damien@chu-lyon.fr (M.D.);; 2Centre de Recherche et d’Innovation en Audiologie Humaine, Institut Pasteur, Institut de l’Audition, rue du Docteur Roux, 75015 Paris, France; 3Faculté de Médecine, Université Claude Bernard Lyon 1, 69003 Lyon, France; 4Service ORL, Centre d’Exploration Fonctionnelle de l’Équilibre chez l’Enfant (EFEE), Hôpital Robert-Debré AP-HP, 75019 Paris, France

**Keywords:** cervical vestibular evoked myogenic potentials, otolith receptors, children, otitis media effusion, bone conduction, air conduction, amplitude ratio, wave latency

## Abstract

Objective: To compare the amplitude ratio and P-wave latency of cervical vestibular evoked myogenic potentials (c-VEMPs) for bone conduction (BC) and air conduction (AC) stimulation in children with otitis media with effusion (OME). Material and methods: This is an observational study of a cohort of 27 children and 46 ears with OME. The c-VEMP amplitude ratio and P-wave latency were compared between BC and AC in children with OME and healthy age-matched children. Results: The c-VEMP response rate in children with OME was 100% when using BC stimulation and 11% when using AC stimulation. The amplitude ratio for BC was significantly higher in the OME group than the age-matched healthy control group (*p* = 0.004). When focusing on ears with an AC c-VEMP response (*n* = 5), there was a significant difference in the amplitude ratio between the AC and BC stimulation modes, but there was no significant difference in the AC results between the OME group and the age-matched control group. Conclusions: BC stimulation allows for reliable vestibular otolith testing in children with middle ear effusion. Given the high prevalence of OME in children, clinicians should be aware that recording c-VEMPs with AC stimulation may lead to misinterpretation of otolith dysfunction in pediatric settings.

## 1. Introduction

Since cervical vestibular evoked myogenic potentials (c-VEMPs) were first described [1,2], measuring c-VEMPs has become a widely used test for the assessment of the otolith function of the peripheral vestibular system. c-VEMPs correspond to the activation of the vestibulospinal pathway and measure the modulation of electromyographic activity of the ipsilateral sternocleidomastoid muscle during an active contraction in response to high-intensity and low-frequency acoustic stimulation delivered by air conduction (AC) or bone conduction (BC) [3,4,5]. This test was initially used for vestibular assessment in adults [6,7]. It has recently been applied to children and is now advised for routine vestibular assessment in children [8,9,10].

In recent years, vestibular function screening has been applied in a variety of clinical contexts. First, congenital or early acquired vestibular disorders should be investigated in cases of any motor delay or repetitive falls; it has indeed been clearly demonstrated that normal vestibular activity is fundamental to the acquisition and appropriate development of balance and postural control [11,12,13]. Then, since the cochlear and vestibular compartments share the same embryological origin and common structural and biomechanical properties, vestibular dysfunction in children can also occur in a genetic environment associated with hearing loss [11,14,15]. According to the literature, the prevalence of vestibular dysfunction in hearing-impaired children varies from 20 to 85% [15,16,17]. Vestibular assessment has also become an essential part of pre-cochlear implant evaluation. Given the potential deleterious effects of implants on vestibular function [18,19,20,21,22], vestibular testing may help prevent and anticipate disabling postoperative bilateral vestibular impairment, which can be difficult to rehabilitate functionally.

The high prevalence of otitis media with effusion (OME) is an important parameter to consider in the otoneurological assessment of children. OME is a frequent condition characterized by the presence of fluid in the middle ear without a sign or symptom of acute ear infection [23,24], spontaneously, in the presence of eustachian tube dysfunction, or after acute otitis media. Approximately 90% of children have at least one episode of OME before school age, usually between 6 months and 4 years of age. In the first year of life, more than 50% of children present with OME. In most cases, OME resolves spontaneously within 3 to 6 months. Considering the high prevalence of OME in children and its potentially significant impact on the conductive properties of the middle ear, the use of AC stimulation for saccular function assessment may have some limitations. Previous studies consistently reported a lower amplitude ratio and absence of c-VEMP response with AC stimulation compared to BC stimulation in adults with conductive hearing loss [25,26,27,28,29]. These results suggest that AC stimulation requires the integrity of the middle ear for optimal transmission of acoustic energy to the saccular system [7].

To our knowledge, no pediatric study has been carried out to assess the value of BC stimulation in children with conductive hearing loss. The aim of this study was therefore to compare AC and BC c-VEMPs in children with OME to normative values in age-matched children.

## 2. Materials and Methods

### 2.1. Population

This study included 27 subjects (46 ears) aged 12 months to 11 years. There were 13 boys and 14 girls. None of the children had a pediatric follow-up since birth that revealed any developmental abnormalities. A neurological and vestibular clinical examination was performed by a skilled neuro-otologist and was normal for all children. At the time of inclusion, OME was confirmed in all participants by an otoscopic examination and type B tympanometry. Subjects were tested for AC and BC c-VEMPs in both ears. Consent forms were obtained from all parents in adherence with the principles of the Declaration of Helsinki, and the study was approved by an ethics committee (Comité de Protection des Personnes, Ile de France; #96048).

### 2.2. Stimulation Parameters

The recording protocol was the same as detailed in a previous study of healthy subjects [30]. All participants were tested for AC and BC c-VEMPs in both ears. The c-VEMP recordings were acquired utilizing a brainstem-evoked auditory response device (Centor plus^®^, Racia Alvar, Bordeaux, France) equipped with an additional amplifier capable of delivering high-intensity-level tone bursts, reaching up to 110 dB HL, equivalent to 96 dB nHL for BC recordings and 98 dB nHL for AC recordings.

For the AC recordings, brief tone bursts (4 ms duration with one cycle each for rise, plateau, and fall times, at a frequency of 750 Hz and a rate of 4 pps) were delivered through headphones (TDH39, Telephonics, Huntington, NY, USA). For the BC condition, a B71 vibrator (RadioEar, New Eagle, PA, USA) was applied to the mastoid on the side corresponding to the tested ear. Calibration of the stimuli followed standard protocols, with AC stimuli being calibrated in dB peSPL (peak-to-peak equivalent SPL) according to the IEC 60645-3:2020 standard [31] and BC stimuli in dB FL.

In the AC condition, TDH39 headphones were calibrated in an artificial ear, conforming to the IEC 60318-1:2009 standard [32], while for BC, the B71 bone vibrator was calibrated on an artificial mastoid, according to the IEC 60318-6:2007 standard [33]. The RETSPL values (reference equivalent threshold for dB nHL scale) for the tone bursts were derived from the ISO 389-6:2007 standard [34]. With this recording system, well-defined biphasic c-VEMP responses with an average of 25 repetitions for each stimulation level were obtained.

Stimulation levels were initiated at 100 dB HL, equivalent to 86 dB nHL for BC and 88 dB nHL for AC. When no response was detected at 86 dB nHL for BC and 88 dB nHL for AC, the stimulation level was increased by 5 dB steps until reaching the maximum output level of the recording system to detect a response. The testing sequence was standardized to begin with AC, and the order of testing between the ears was randomized. No data were excluded due to reliability issues, and the entire testing protocol was successfully completed for all participants. A visual representation of BC cVEMP recordings is shown in Figure 1.

### 2.3. Electromyographic and c-VEMP Recordings and Characterization

The c-VEMP recordings were executed using active surface electrodes strategically positioned along the anterior margin of the sternocleidomastoid muscle. The reference electrode was meticulously placed at the suprasternal notch, while the grounding electrode was thoughtfully situated in the middle of the forehead at the nasion. The EMG was averaged for each tone burst and individual trace within a 150-milisecond window starting at the onset of the tone burst. Subsequently, the peak-to-peak c-VEMP response was examined within a 50-milisecond window, beginning with the onset of the stimulus. To ensure consistency and reliability in our analysis, all traces collected at the same EMG level during a recording session were methodically selected for offline averaging. This systematic approach facilitated comparisons of c-VEMP parameters such as amplitude, latency, and threshold both within the same recording session and across sessions conducted at distinct time intervals. To minimize the effects of inter- and intra-individual variability in muscle contractions, the c-VEMP peak-to-peak amplitude was expressed relative to the EMG level as an amplitude ratio (PN/EMG) and measured at the reference stimulation level.

In each recording session, c-VEMP responses underwent analysis for several parameters, including amplitude (P-to-N peak amplitude, P-N), latency (P latency for the first peak and N latency for the second peak), and threshold.

Inter-aural relative asymmetry was determined by calculating the c-VEMP amplitude ratio using the following formula: ([Right ear − Left ear]/[Right ear + Left ear] × 100).

### 2.4. Test Set-Up

The optimal posture for inducing sustained neck muscle contractions was determined based on the age of the participants. The youngest children were seated on their parent’s lap, face-to-face, with the parent instructed to provide support to the lower part of the child’s back. The examiner gently tilted the child backward, turning their head away from the side where the stimulation was applied. This positioning encouraged the child to straighten up and focus on or reach for a toy placed on that side.

Children aged 6 years and older were positioned in a partially reclined manner on an examination table and were instructed to turn their heads away from the stimulated ear. To ensure precision in assessing neck muscle contractions, both the patient and the examiner had access to real-time visual feedback from the EMG recording, displayed as a car speedometer on a monitor. This allowed for continuous monitoring and verification of the patient’s muscle contractions. The target range for EMG activity was set at 100–500 μV.

During data acquisition, recording was terminated as soon as consistent c-VEMP responses were achieved, typically requiring 25 repetitions. In cases where reproducible responses were not obtained at the maximum stimulation level, the number of acquisitions was systematically increased to 50. For validation purposes, recordings at threshold were replicated at least twice.

To facilitate cross-subject comparisons, responses to stimulations at 100 dB HL (equivalent to 86 dB nHL for BC and 88 dB nHL for AC) were chosen as the reference values, as these yielded clear c-VEMP parameters for nearly all patients. This stimulation level was replicated twice for validation.

### 2.5. Statistical Analyses

We first analyzed the AC and BC stimulation response rates in the 46 ears with OME. Then, BC c-VEMP characteristics (amplitude ratio and P-wave latency) were compared to those of age-matched controls. The control subjects were selected from a recent observational study by our group reporting age- and sex-specific normative values for c-VEMPs in children aged 6 months to 15 years [30]. We were able to match 40 of the 46 ears included in this study according to age and sex. Since the data for each group did not follow a normal distribution, we used the non-parametric Mann–Whitney test for independent samples.

Additional statistical analyses were carried out based on the descriptive results. Among the ears for which AC c-VEMP responses were obtained, we first compared the AC c-VEMP characteristics (amplitude ratio and P-wave latency) with the BC c-VEMP characteristics for the same ears. To that end, we used a paired *t*-test after confirming the normal distribution of the data using a Shapiro–Wilk test and the homogeneity of the variances using the Fisher F test. Then, we compared the AC c-VEMP characteristics (amplitude ratio and P-wave latency) with the AC c-VEMP characteristics of the age-matched controls. To that end, we used a non-parametric Mann–Whitney test as the data were not normally distributed. For all the descriptive and statistical analyses, we used the XLSTAT program for Mac, with a significance level of 0.05.

## 3. Results

In the OME group, 41 ears (89%) showed no AC c-VEMP response at 110 dB HL (98 dB nHL), while all children (100%) showed normal BC c-VEMP responses.

### 3.1. Comparison 1

We first compared the BC c-VEMP characteristics in the OME group with those of the age-matched control group. Analyses showed no significant difference in the P-wave latency (Mann–Whitney test, *p* = 0.381) between children with OME and age-matched controls. A significant difference (Mann–Whitney test, *p* = 0.004) in the amplitude ratio was found between the two groups, with a higher mean value in the OME group (Figure 2). The BC c-VEMP amplitude ratio for each ear was then compared to age- and sex-specific normative values [30]; 50% of the ears with OME had an amplitude ratio above the 95-percentile of the normative values.

### 3.2. Comparison 2

We identified AC c-VEMP responses in 5 of the 46 ears evaluated. Then, the c-VEMP responses in both stimulation modes were analyzed. The AC amplitude ratios were compared with the BC amplitude ratios in the same ears (paired *t*-test, *p* = 0.019) and with the AC amplitude ratios of age-matched controls (Mann–Whitney test, *p* = 0.310) (Figure 3). P latencies were also compared, with no significant difference.

All detailed results are presented in Table 1, Table 2 and Table 3.

## 4. Discussion

The present study investigated the value of BC c-VEMPs versus AC c-VEMPs in children with OME. The results showed a sensitivity of 100% for BC stimulation, compared to only 11% for AC stimulation. Consistent with our results, Bath et al. [26] successfully elicited AC c-VEMP responses in only 2 of 23 adult ears (8%) with conductive hearing loss. Han et al. [35] showed a maximal sensitivity of AC c-VEMPs of 100% that moderately decreased to 61.9% when simulating conductive hearing loss (using a soundproof earplug) with an air–bone gap of 22 dB. In 21 adults with middle ear effusion, Wang et al. [36] found an AC c-VEMP response rate of 67% with a significant increase at 95% after tympanic aspiration. El Kousht et al. [28] showed a maximum response rate in both stimulation modes in 10 healthy adult subjects (20 ears) and 10 patients with sensorineural hearing loss (20 ears). In a third group with conductive hearing loss (20 ears), the AC response rate fell to 60%. All these results in adult patients support the idea that AC c-VEMP sensitivity is vulnerable to middle ear conduction impairment. The most plausible explanation is that OME alters the impedance of the tympano-ossicular system and reduces the transmission of acoustic energy to the inner ear and saccular component, making stimulation ineffective in activating otolith receptors. Similarly, Merchant et al. [37] suggested that the abolition of the AC c-VEMP response observed after cochlear implantation could be due to induced mechanical changes resulting from an air–bone gap. The authors showed that among 26 ears with cochlear implants, 10 ears (38%) had c-VEMP responses for both AC and BC, 8 ears (31%) had responses only for BC, and 8 ears (31%) had no response. The c-VEMP response rates for AC and BC after cochlear implantation were 41% and 67%, respectively. Therefore, the absence of an AC response in ears with cochlear implants may not systematically correspond to vestibular loss and justifies BC testing.

### 4.1. BC c-VEMP Characteristics of the OME Group Compared to the Control Group

In the present study, no significant difference was found between OME subjects and age-matched controls regarding P-wave latency for BC c-VEMPs. However, a significant difference in the amplitude ratio between the OME group and the age-matched control group was observed, with a higher mean value in the OME group. These results suggest that the presence of middle ear effusion may tend to increase the response amplitude of the saccular system to BC acoustic stimulation. Hypothetically, based on the present results, we assumed that this larger sacculocolic response induced by BC could be explained by the same acoustic principles involved in the Weber phenomenon. Skull vibrations activate the cochlea by several biophysical mechanisms, such as distortional, inertial, and compressional mechanisms, that all contribute to generating the “traveling wave” described by Bekesy [38]. In a recent paper, the authors showed that distortion on the basilar membrane induced by BC stimulation also produces an otoacoustic emission toward the oval window and the middle ear [39].

As discussed above, the middle ear plays an impedance-matching role to minimize the loss of acoustic energy due to the difference in impedance between the surrounding air and the labyrinthine fluids. The presence of middle ear effusion could modify the biomechanical properties of this impedance-matching system, i.e., the reflection coefficient at the oval window, prevent external dissipation of an acoustic signal, and lead to increased cochlear stimulation and an increased perception of sound. In an analogous way, the present findings could be explained by the same acoustic principle, resulting in increased saccular stimulation.

Another explanation of our findings can be attributed to the concept of “resonance frequency”, which is defined as the natural frequency at which a physical or mechanical system is most responsive. The resonant frequency of the temporal bone plays a pivotal role in the transmission of sound waves to the inner ear. In our OME group, the presence of effusion in the tympanic cavity may have affected the temporal density and thus lowered the system’s resonance frequency [40]. According to laws of physics and acoustics, as the stimulation frequency approaches the resonance frequency, the amplitude of vibration of the temporal bone and the inner ear can increase and enhance the transmission of auditory and vestibular inputs.

Unfortunately, the literature is relatively poor, and few studies discuss the difference in BC c-VEMP responses between healthy subjects and patients with conductive hearing loss. Inconsistent with the present results, El Kousht et al. [28] did not find any significant difference when comparing BC c-VEMP amplitude responses in healthy adults and subjects with conductive hearing loss (*n* = 20 ears).

### 4.2. Comparative Analysis Based on c-VEMP Characteristics in the OME Group with AC Responses

In the second part of the present work, we focused on the five ears that presented c-VEMP responses in both stimulation modes. We compared the AC and BC responses in the OME group and the AC responses between the OME group and the age-matched control group. The results showed a significantly greater amplitude ratio for BC compared to AC in the OME group, which can be explained by the same arguments mentioned in the first paragraph. The other results are consistent with those reported by El Kousht et al. [28], who did not find any statistically significant difference between healthy adults (*n* = 20 ears) and those with conductive hearing loss (*n* = 20 ears) regarding the amplitude ratio (*p* = 0.876) or P13-wave latency (0.437) for the AC condition. Conversely, Han et al. [35] and Wang et al. [36] demonstrated significantly longer latencies in subjects with conductive hearing loss.

### 4.3. Limitations

While BC stimulation serves as a valuable alternative in many clinical scenarios, this test does not provide a strictly unilateral assessment of the saccular system in a purely theoretical sense. Indeed, acoustic transcranial transmission properties may explain the absence of inter-aural attenuation of BC, which consequently stimulates both ears equally. However, as c-VEMP records the electromyographic response of the ipsilateral sternocleidomastoid muscle and BC stimulates both ears equally, questions remain about the involvement of the contralateral vestibule in the ipsilateral vestibulospinal response. Research conducted by Colebatch et al. [2] demonstrated that acoustic stimulation primarily elicits ipsilateral sacculospinal projections to the SCM muscle. However, there is evidence of contralateral vestibular input crossing over to the ipsilateral SCM muscle. Nonetheless, these projections are weak, infrequent, and excitatory in nature [41]. Some authors have even suggested a utricular origin for these contralateral projections, implying the absence of sacculospinal crossover projections [42].

Therefore, the presence of a contralateral vestibulospinal pathway to the ipsilateral SCM muscle (albeit weak and infrequent) could, to some extent, interfere with the ipsilateral sacculospinal response. Hence, it is important to emphasize that there is currently no clinical evidence supporting this occurrence in the literature. Furthermore, these crossed vestibulospinal projections are very sparsely represented, suggesting a weak to negligible effect on a unilateral sacculospinal response.

Another significant aspect to consider is that responses may differ in terms of polarity and/or latency depending on the stimulation position [43,44,45]. While the absence of inter-aural attenuation has been confirmed using a forehead vibrator, there are inconsistent results suggesting that some degree of inter-aural attenuation may still be present when using mastoid stimulation [40]. Therefore, the acoustic energy perceived by the contralateral ear may not be clinically sufficient, considering the high threshold of acoustic stimulation of the otolith system.

In conclusion, it is worth noting that even though BC stimulation theoretically affects both ears, the contribution of the contralateral vestibule to the ipsilateral cVEMP response is minimal, if not negligible.

## 5. Conclusions

The present study shows that in children with a middle ear effusion, BC stimulation is more reliable than AC stimulation for assessing otolith function. These results could be extended to all conductive disorders (e.g., ossicular chain dislocation, juvenile otosclerosis). Recording cVEMPs using BC stimulation should help avoid misinterpretations of otolith dysfunction in children with OME, a very common pediatric condition.

## Figures and Tables

**Figure 1 jcm-12-06348-f001:**
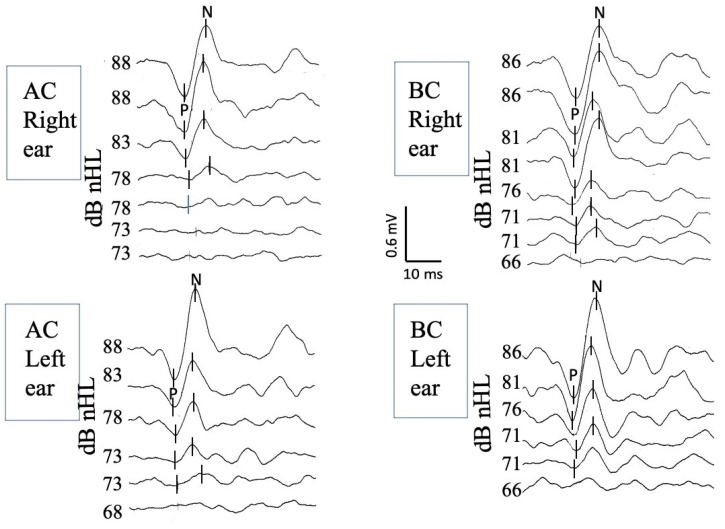
BC cVEMP recordings of the right ear (top traces) and the left ear (bottom traces) obtained from a 4-year-old girl. Well-identified biphasic c-VEMP responses are observed bilaterally at decreasing stimulation intensities up to the response threshold. The latency of the first peak is measured at approximately 13 ms (P wave, also known as P13), and the latency of the second at approximately 23 ms (N wave, also known as N23). dB nHL: decibels normalized hearing level; BC: bone conduction; cVEMPs: cervical vestibular evoked myogenic potentials.

**Figure 2 jcm-12-06348-f002:**
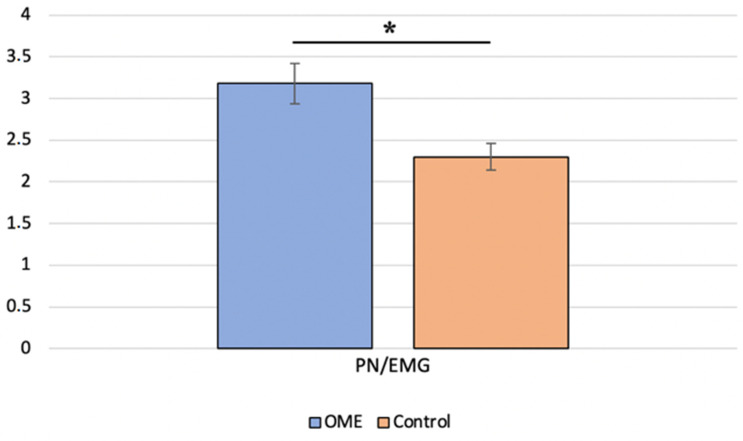
Comparison of the P-to-N amplitude ratio for bone conduction between children with otitis media with effusion (OME group) and age-matched healthy controls (control group) * indicates a significant difference.

**Figure 3 jcm-12-06348-f003:**
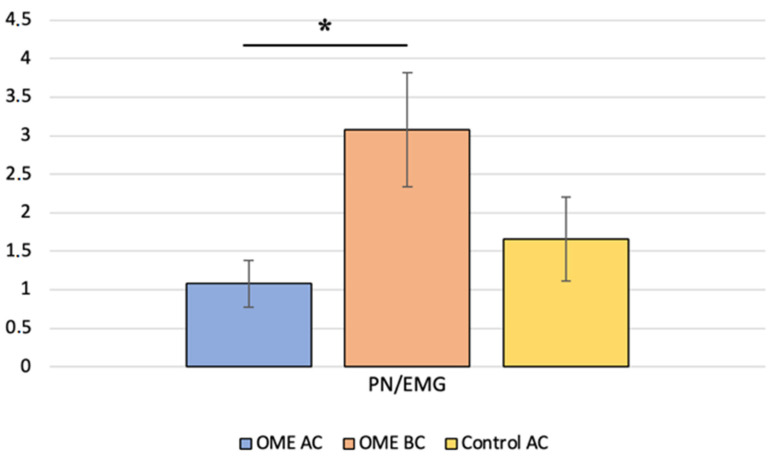
Comparison of the P-to-N amplitude ratio between children with otitis media with effusion (OME group) for air conduction (AC) and bone conduction (BC) and with age-matched healthy controls (control group) for AC; * indicates a significant difference.

**Table 1 jcm-12-06348-t001:** Comparison of the P13 latency and amplitude ratio for bone conduction (BC) stimulation between children with otitis media with effusion (OME) and age-matched controls (Control). EMG: electromyography. * indicates a significant difference between two values.

	P13 Latency (ms)		Amplitude Ratio P-N/EMG	
Group	OME (*n* = 40)	Control (*n* = 40)	OME (*n* = 40)	Control (*n* = 40)
Mode	BC	BC	BC	BC
Mean ± SD	12.72 ± 0.63	12.83 ± 0.78	3.18 ± 1.53	2.3 ± 1.03
Median [IQR]	12.6 [12.4–13.1]	12.7 [12.4–13.2]	3.5 [2.1–4.2]	2.3 [1.7–3.0]
SEM	0.10	0.12	0.24	0.16
*p* value	0.381		0.004 *	

**Table 2 jcm-12-06348-t002:** Comparison of the P13 latency and amplitude ratio between air conduction (AC) and bone conduction (BC) stimulation in children with otitis media with effusion. EMG: electromyography. * indicates a significant difference between two values.

	P13 Latency (ms)		Amplitude Ratio P-N/EMG	
Mode	AC	BC	AC	BC
Mean ± SD	12.47 ± 0.83	12.76 ± 0.65	1.08 ± 0.66	3.08 ± 1.64
Median [IQR]	13 [12.2–13.4]	12.4 [12.4–13]	1.00 [0.7–1.6]	4.00 [2.6–4.1]
SEM	0.37	0.29	0.30	0.73
*p* value	0.646		0.019 *	

**Table 3 jcm-12-06348-t003:** Comparison of the P13 latency and amplitude ratio for air conduction (AC) stimulation between children with otitis media with effusion (OME) and age-matched controls (Control). EMG: electromyography.

	P13 Latency (ms)		Amplitude Ratio P-N/EMG	
Group	OME (*n* = 5)	Control (*n* = 5)	OME (*n* = 5)	Control (*n* = 5)
Mode	AC	AC	AC	AC
Mean ± SD	12.47 ± 0.83	12.76 ± 1.35	1.08 ± 0.66	1.66 ± 1.24
Median [IQR]	13 [12.2–13.4]	12.2 [12–13.4]	1.00 [0.7–1.6]	0.90 [0.7–2.7]
SEM	0.37	0.60	0.30	0.55
*p* value	0.810		0.310	

## Data Availability

The data presented in this study are available on request from the corresponding author. The data are not publicly available due to ethical, legal, and privacy issues.

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
