# Peer review of "Bone Conduction Cervical Vestibular Evoked Myogenic Potentials as an Alternative in Children with Middle Ear Effusion"

_jcm, 2023, doi:10.3390/jcm12196348_

Round 1
Reviewer 1 Report
I suggest to complete the following in Materials and Methods:
- the position of the child/head of the child along the cVEMP recordings
- the position of all electrodes to record the cVEMPs , not only the active electrode.
Although the authors mention that The recording protocol was the same as detailed in a previous study of healthy sub-87 jects [28]. , I suggest to mention briefly the above details into this article too.
Author Response
The entire protocol is explained and detailed as you suggested.
Reviewer 2 Report
- As the authors noted, the clinical value of BC c-VEMP in children with conductive hearing loss has been sparsely investigated. Thus, this study is worthwhile, and the methodology was appropriate; however, it is unfortunate that there is nothing new in the results.
- In clinical practice in pediatrics, checking the status of the tympanic membrane via otoscopy is a must, because, as the authors mentioned, otitis media with effusion (OME) is quite common. Not only in c-VEMP, but also in evoked otoacoustic emissions that assess hearing, middle ear can affect the results, so it is imperative to check the status of middle ear. It is not suprising the low sensitivity in AC c-VEMP in Figure 2.
- The idea that we can do c-VEMP via bone conduction stimulation in patients with OME is not new, and many clinicians do that if a patient with OME really needs to be evaluated via c-VEMP. However, I think it is very rare to do BC c-VEMP in a child who presents with both dizziness and OME. If a child with dizziness comes in, I check the tympanic membrane. If the child has OME, I may not do c-VEMP to assess vestibular function. Rather, I’ll do a tympanostomy to drain the effusion to ventilate the middle ear and hope for symptomatic improvement. If the effusion resolves and the dizziness persists, then I might consider c-VEMP, but I would not necessarily do BC c-VEMP at this point, but AC c-VEMP.
- BC c-VEMP would be indicated in children with a number of other conductive hearing loss, even if they do not have OME. However, because the BC stimulates both ears simultaneously, it is difficult to assume that the measured response purely reflects the function of unilateral saccule and vestibulospinal tract (Halmagyi GM, Curthoys IS, Colebatch JG, Aw ST. Vestibular responses to sound. Ann N Y Acad Sci. 2005;1039:54-67. doi:10.1196/annals.1325.006). This point deserves some discussion, but nothing was mentioned. The higher amplitude ratio measured in the BC c-VEMP may also be due to this matter.
- Figures are nothing more than a repetition of the Tables, so it does not seem necessary.
The quality of English is fine.
Author Response
- As the authors noted, the clinical value of BC c-VEMP in children with conductive hearing loss has been sparsely investigated. Thus, this study is worthwhile, and the methodology was appropriate; however, it is unfortunate that there is nothing new in the results.
Recently, we published a study in a large cohort of healthy children (n=137) comparing c-VEMP results using the 2 modes of stimulation, air conduction and bone conduction. The main objective was to establish normative values according to age, gender and the type of stimulation used. As a complement of this published data, we decided to extend this study to children with OME and compare the results with age-matched healthy children, as no prior research was found on this specific topic.
- In clinical practice in pediatrics, checking the status of the tympanic membrane via otoscopy is a must, because, as the authors mentioned, otitis media with effusion (OME) is quite common. Not only in c-VEMP, but also in evoked otoacoustic emissions that assess hearing, middle ear can affect the results, so it is imperative to check the status of middle ear. It is not surprising the low sensitivity in AC c-VEMP in Figure 2.
We agree that cVEMP are not used for checking the status of the tympanic membrane and identify middle ear effusion (also otoscopy may be quite often in default to assess or refute OME).
The aim of acoustic stimulation by air or bone conduction is to provide VEMP responses that reflect otolith vestibular descending pathways in clinically indicated conditions: dizziness complains and balance disorders, delayed posturomotor development, congenital hearing loss, pre-and post cochlear implant assessment. These tests allow us to exclude vestibular disorders, or pathology with balance problems and vertigo of not vestibular origin (for exemple migraine or ophthalmologic disorders-
It is effectively not surprising that middle ear effusion decreases VEMP responses with AC stimuli like in other conductive disorder (blockage of the ossicular chain or inflammatory material after cochlear implant).
This study is therefore intended to raise awareness in otoneurologists to consider the bone conduction c VEMP test in pediatric populations when there is no response to air conduction stimulation and/or in cases of doubt regarding the middle ear status.
Finally, OAE are not a good system to assess for middle ear effusion since this test may be altered in any kind of hearing loss (conductive and sensorineural) > 30 dBHL.
- The idea that we can do c-VEMP via bone conduction stimulation in patients with OME is not new, and many clinicians do that if a patient with OME really needs to be evaluated via c-VEMP. However, I think it is very rare to do BC c-VEMP in a child who presents with both dizziness and OME. If a child with dizziness comes in, I check the tympanic membrane. If the child has OME, I may not do c-VEMP to assess vestibular function. Rather, I’ll do a tympanostomy to drain the effusion to ventilate the middle ear and hope for symptomatic improvement. If the effusion resolves and the dizziness persists, then I might consider c-VEMP, but I would not necessarily do BC c-VEMP at this point, but AC c-VEMP.
As you mentioned earlier, OME is a very common condition in children. Nevertheless, the presence of OME should not delay or interfere with a complete vestibular evaluation. As mentioned in our article, OME is a very common condition, but it is also generally spontaneously resolving. The indication for tympanostomy would appear to be an unnecessary procedure, given that bone conduction can bypass middle ear conduction problems and show maximum sensitivity.
- BC c-VEMP would be indicated in children with a number of other conductive hearing loss, even if they do not have OME. However, because the BC stimulates both ears simultaneously, it is difficult to assume that the measured response purely reflects the function of unilateral saccule and vestibulospinal tract (Halmagyi GM, Curthoys IS, Colebatch JG, Aw ST. Vestibular responses to sound. Ann N Y Acad Sci. 2005;1039:54-67. doi:10.1196/annals.1325.006). This point deserves some discussion, but nothing was mentioned. The higher amplitude ratio measured in the BC c-VEMP may also be due to this matter.
Indeed, we may add a paragraph to mention this.
“While bone BC stimulation serves as a valuable alternative in many clinical scenarios, this test does not provide a strictly unilateral assessment of the saccular system in a purely theoretical sense. Indeed, acoustic transcranial transmission properties may explain the absence of interaural attenuation in BC, which consequently stimulates both ears equally. However, as c-VEMP records the electromyographic response of the ipsilateral sternocleidomastoid muscle and BC may stimulates both ears equally, questions remain about the involvement of the contralateral vestibule to the ipsilateral vestibulospinal response. The research conducted by Colebatch et al (2) demonstrated that acoustic stimulation primarily elicits ipsilateral saccular-spinal projections to the SCM muscle. However, there is some evidence of contralateral vestibular input crossing over to the ipsilateral SCM muscle. Nonetheless, these projections are weak, infrequent, and excitatory in nature (36). Some authors even suggested an utricular origin for these contralateral projections, implying the absence of sacculo-spinal crossover projections (37).
Therefore, the presence of a contralateral vestibulo-spinal pathway to the ipsilateral SCM muscle (albeit weak and infrequent) could, to some extent, interfere with the ipsilateral saccular-spinal response. Hence, it is important to emphasize that there is currently no clinical evidence supporting this occurrence in the literature. Furthermore, these crossed vestibulospinal projections are very sparsely represented, suggesting a weak to negligible effect on unilateral sacculospinal response.
Another significant aspect to consider is that responses may differ in terms of polarity and/or latency depending on the stimulation position (39–41). While the absence of interaural attenuation has been confirmed using a forehead vibrator, there are inconsistent results suggesting that some degree of interaural attenuation may still be present when using mastoid stimulation (38). Therefore, the acoustic energy perceived by the contralateral ear may not be clinically sufficient, considering the high threshold of acoustic stimulation of the otolith system.
In conclusion, it is worth noting that even though BC stimulation theoretically affects both ears, the contribution of the contralateral vestibule to the ipsilateral cVEMP response is minimal, if not negligible”.
- Figures are nothing more than a repetition of the Tables, so it does not seem necessary.
We can agree that the figures, even if they illustrate the results, contribute redundancy and may be considered unnecessary.

Round 2
Reviewer 2 Report
The authors have done a good job of defending the value of this study. The authors' explanation for my review points were helpful. The part where I asked for further discussion is also well described.